# A Randomized Controlled Trial of Soy Isoflavone Intake on Mammographic Density among Malaysian Women

**DOI:** 10.3390/nu15020299

**Published:** 2023-01-06

**Authors:** Nadia Rajaram, Beverley Yap, Mikael Eriksson, Shivaani Mariapun, Lee Mei Tan, Hamizah Sa’at, Evelyn Lai Ming Ho, Nur Aishah Mohd Taib, Geok Lin Khor, Cheng Har Yip, Weang Kee Ho, Per Hall, Soo Hwang Teo

**Affiliations:** 1Cancer Research Malaysia, Subang Jaya 47500, Malaysia; 2Karolinska Institutet, 171 77 Stockholm, Sweden; 3University of Malaya Cancer Research Institute, Kuala Lumpur 50603, Malaysia; 4ParkCity Medical Centre, Ramsay Sime Darby Healthcare, Kuala Lumpur 52200, Malaysia; 5Department of Nutrition and Dietetics, Universiti Putra Malaysia, Serdang 43400, Malaysia; 6Subang Jaya Medical Centre, Ramsay Sime Darby Healthcare, Subang Jaya 47500, Malaysia; 7Department of Applied Mathematics, Faculty of Engineering, University of Nottingham Malaysia, Semenyih 43500, Malaysia; 8Södersjukhuset, 118 83 Stockholm, Sweden

**Keywords:** Soy isoflavones, mammographic density, randomized controlled trial, breast cancer

## Abstract

Soy intake is associated with lower breast cancer risk in observational studies concerning Asian women, however, no randomized controlled trials (RCT) have been conducted among Asian women living in Asia. This three-armed RCT assessed the effects of one-year soy isoflavone (ISF) intervention on mammographic density (MD) change among healthy peri- and postmenopausal Malaysian women. This study was registered at ClinicalTrials.gov (NCT03686098). Participants were randomized into the 100 mg/day ISF Supplement, 50 mg/day ISF Diet, or control arm, and assessed for change in absolute and relative dense area from digital mammograms conducted at enrolment and after 12 months, compared over time across study arms using Kruskal-Wallis tests. Out of 118 women enrolled, 91 women completed the intervention, while 27 women (23%) were lost in follow up. The ISF supplement arm participants observed a larger decline in dense area (−1.3 cm^2^), compared to the ISF diet (−0.5 cm^2^) and control arm (−0.8 cm^2^), though it was not statistically significant (*p* = 0.48). Notably, among women enrolled within 5 years of menopause; dense area declined by 6 cm^2^ in the ISF supplement arm, compared to <1.0 cm^2^ in the control arm (*p* = 0.13). This RCT demonstrates a possible causal association between soy ISF intake and MD, a biomarker of breast cancer risk, among Asian women around the time of menopause, but these findings require confirmation in a larger trial.

## 1. Introduction

Globally, breast cancer continues to be the most common cancer in women and a leading cause of cancer-related deaths [1]. Asian women are experiencing a rapid rise in incidence, by up to a 6% increase annually [2], that has been attributed to changes in reproductive factors and westernization of lifestyles [3,4,5]. Therefore, there is an urgent need to identify effective primary prevention strategies that could reduce the number of Asian women diagnosed with breast cancer in the near future. Drugs that are aromatase inhibitors and selective estrogen receptor modulators, such as tamoxifen and raloxifene, are actively studied for primary prevention of breast cancer in populations with higher incidence of breast cancer [6], but the risk of side effects could outweigh the benefits among Asian women with lower risk.

Dietary interventions, such as soy intake, may be an important strategy for breast cancer risk reduction in this region. Asian populations with historically lower breast cancer incidence have soy-rich diets, up to 10-fold more compared to Caucasian women [7]. Soy foods are rich in isoflavones (ISFs), which are natural phytoestrogens and selective estrogen receptor modulators [8]. ISFs are hypothesized to compete with endogenous estrogen to selectively bind to estrogen receptor sites. However, they exert a weaker effect compared to endogenous estrogen, and therefore, reduce estrogen exposure [8,9]. Studies have demonstrated that soy isoflavones may be equally as effective as hormone replacement therapy in minimizing conditions association with the menopausal transition, such as osteoporosis and somatic-vegetative symptoms [10,11]. Observational studies of Asian women have consistently reported that high soy intake is associated with between 14 and 41% lower relative risk of breast cancer [7,12,13,14].

To date, there are no intervention studies of soy isoflavones with breast cancer occurrence as the primary endpoint because it requires large cohorts followed over long periods of time, making such studies expensive and difficult. Instead, randomized controlled trials (RCTs) have tested interventions of 80–120 mg of soy isoflavones on biomarkers of breast cancer risk among Caucasian women, such as mammographic density (MD), but have consistently reported null or small effects [15,16,17,18,19,20,21]. MD represents the area or volume of fibro-glandular tissue in the breast that is radiologically visible on a mammogram image. In a meta-analysis of RCTs that use MD as a biomarker of breast cancer risk, high doses of isoflavones were associated with a small, non-significant protective effect, particularly among post-menopausal women [15].

There are a few hypotheses that were proposed to explain the lack of a clinically meaningful effect in previous RCTs. First, the RCTs used purified and concentrated soy ISF supplements instead of traditionally consumed soy foods made from whole soybeans [22]. Second, the protective effect seen among Asian women may be due to lifelong exposure to soy, particularly in adolescence [16,18,19,23]. Furthermore, there may be population differences in the ability to metabolize soy ISFs into potentially more potent metabolites [24,25]. These hypotheses suggests that the causal association between soy ISF intake and breast cancer risk may be elucidated in a study of Asian women living in Asia.

Thus, we conducted a randomized controlled trial to determine if a soy ISF intervention, either consuming 100 mg/day of soy isoflavone supplement or 50 mg/day of soy ISFs through dietary sources of soy, could reduce breast cancer risk among peri- and post-menopausal Malaysian women, using MD as a biomarker of risk. To our knowledge, this is the first interventional trial of soy isoflavone intake, using both dietary and supplement sources of isoflavones, on breast cancer risk among Asian women living in Asia where soy foods are accessible, affordable, and palatable.

## 2. Materials and Methods

### 2.1. Study Participants

The Malaysian Soy and Mammographic Density (MiSo) Study is a three-armed, randomized controlled trial with no blinding and no placebo control. Peri- and postmenopausal Malaysian women between 45 and 65 years old were invited to participate in the trial between November 2018 and December 2019 at a private tertiary hospital in Malaysia (Subang Jaya Medical Centre). Participants were recruited from an established research database of women attending mammography screening at Subang Jaya Medical Centre and University Malaya Medical Centre, as well as by opportunistic recruitment via social media. All procedures performed in studies involving human participants were in accordance with the ethical standards of the institutional research committees and with the 1964 Helsinki declaration and its later amendments or comparable ethical standards. The study protocol and source documents were approved by the Ramsay Sime Darby Independent Ethics Committee (Reference number: 201805.1), and the study is registered with the Malaysian National Medical Research Register (NMRR-18-287-40385) and ClinicalTrials.gov (NCT03686098). All participants signed the informed consent form prior to any study procedures.

A total of 177 women signed the informed consent form and were screened for eligibility (Figure 1), of whom 3 women did not complete eligibility screening and 56 women reported one or more exclusion criteria. Women were excluded from the study if they reported a menstrual period in the past 3 months (premenopausal) or use of hormone replacement therapy in the past 6 months (*n* = 10). They were excluded for medical conditions including a history of cancer (*n* = 1); current management of gout, diabetes, or hyperthyroidism (*n* = 17); history of gastrointestinal disorders or intolerance to soy (*n* = 5); or if they reported a history of benign breast disease or presented with abnormal mammogram findings during eligibility screening (*n* = 18). Women were also excluded if they were screened by mammography in the past 12 months (*n* = 5) or if they reported daily intake of any type of soy foods (excluding condiments such as soy sauce) or soy-based supplements (*n* = 5). Upon enrollment, participants were assigned to an intervention arm using a digitally-generated randomization list, in which participants were randomized based on the order in which they enrolled. The study manager and study coordinator generated the random allocation sequence, enrolled, and assigned participants to their intervention. We used a stratified, block randomization approach to account for potential differences in distribution by ethnicity and menopausal status. A total of 118 women were randomized to receive either a daily soy ISF supplement (100 mg of soy ISFs daily), a high soy diet (50 mg of soy ISFs daily), or no addition to their diet (control arm).

### 2.2. Data Collection

A structured questionnaire was used to collect information about participant characteristics at enrolment, including socio-demographic profiles, reproductive factors, physical activity, and medical history. The questions were adapted from a questionnaire previously used in this study population, as described elsewhere [26]. The frequency and intensity of physical activity reported in the structured survey was converted into metabolic equivalent of task (MET) hours/week, using a previously reported methodology in this population [27]. Height and weight were measured using a floor scale and were used to calculate body mass index (kg/m^2^). Waist and hip circumference were measured using a standard measuring tape and were used to calculate waist-to-hip ratio.

To assess for dietary changes over the study period, a semi-quantitative food frequency questionnaire (FFQ) was adapted from a questionnaire validated for use among Malaysians [28]. Using the Malaysian and Singaporean Food Composition databases, information about frequency and portion of intake was converted to daily calorie intake (kcal/day) as well as daily protein, fat, carbohydrates [29,30], and soy ISF intake [31]. Compliance and adverse events were assessed monthly using phone calls, online surveys, and study site visits.

### 2.3. Intervention and Compliance

Women in the ISF supplement arm were asked to consume 2 soy ISF supplement tablets once daily. Each tablet of the supplement contained 125 mg of soybean (Glycine max) standardized extract that delivered 50 mg of ISFs (consisting of 92% daidzein and 8% genistein in aglycone weight). Women in the ISF diet arm were asked to consume two daily servings of soy foods, and each serving delivered approximately 25 mg of ISFs. Women in the ISF diet arm were provided with a food guide containing serving sizes of common soy foods, as well as a cash subsidy of RM95/month to purchase soy foods. Women in the control arm were asked to continue their regular diet and were advised to limit foods rich in soy ISFs to a maximum of three servings per week, consistent with regular intake in this cohort [26]. Compliance in the ISF supplement arm was determined by tablet count and was defined as intake of ≥80% of allocated supplements. Compliance in the ISF diet arm and control arm were determined from dietary soy isoflavone intake self-reported in the FFQ and were defined as intake of ≥40 mg/day and <40 mg/day of soy ISFs, respectively.

### 2.4. Outcome Measurements

The primary outcome in this trial was change in mammographic density (MD) over the study period. Mammography was conducted at enrolment and at Month 12 at the study site, using a single full-field digital mammography machine (Hologic Selenia Mammography System, Hologic Inc., Marlborough, MA, USA). MD was estimated from processed digital mammogram images using Stratus, a fully-automated software that is described elsewhere [32]. To standardize for possible differences in breast position during imaging, all mammogram images for each woman were aligned prior to MD estimation [32]. We reported dense area (defined as the total area of fibro-glandular tissue or radiodensity observable on the mammogram image, cm^2^) and percent density (dense area ÷ total breast area × 100%). All MD measures were an average of MD measured for left and right breasts and for both view positions (craniocaudal and mediolateral). The absolute MD change was calculated by subtracting MD estimated at enrolment from MD estimated at Month 12. The relative MD change was defined as the absolute change divided by MD estimated at enrolment and multiplied by 100%.

To ease comparison to the published literature, we also calculated absolute and relative MD change using dense volume estimates from a commonly used mammographic density measurement software (Volpara^TM^). It is important to note that image alignment was not possible for Volpara^TM^ analyses.

### 2.5. Statistical Analysis

Sample size calculations estimated that a total enrolment of 228 women (76 per arm) were required to detect a 2.5% difference in percent density change between any two arms at 80% power and considering a 5.0% standard deviation [33]. Bonferroni correction for multiple comparisons was also considered. This study reached 52% of the enrolment target.

Standard descriptive statistics were used to describe differences across the study arms. Fisher’s Exact tests were used for categorical and ordinal variables. Due to small sample sizes and skewed distributions, Kruskal-Wallis tests were used to test for differences across study arms for continuous variables. Wilcoxon-rank-sum tests were used to test for within-women changes over time.

Kruskal-Wallis tests were also used to compare mammographic density change across study arms. The primary analysis included all women who completed the study (*n* = 90). The per-protocol analysis was restricted to women who were compliant with the intervention (*n* = 68). Sensitivity analyses were conducted to assess the effect of (1) excluding women with low MD at enrolment (lowest 25th percentile of the distribution), (2) excluding women with high habitual soy isoflavone intake at enrolment (highest 25th percentile of the distribution), and (3) isoflavone dosage, defined as quartiles of total reported soy isoflavone intake at Month 12 (including from diet and supplements). The primary analysis was further stratified to assess the effect of time since menopause (<5 vs. ≥5 years) on MD change across the study arms.

All hypothesis testing was two-sided, and a *p*-value of <0.05 was considered statistically significant. All statistical tests were conducted in the R statistical environment, v4.0.3.

## 3. Results

### 3.1. Follow Up during the Study Period

In this randomized controlled trial to investigate the association between soy isoflavone intake and mammographic density as a biomarker of breast cancer risk, 118 women were enrolled and randomized into three study arms (Figure 1). With an average follow up period of 13.3 months, a total of 91 women (77.1%) completed the study (Table 1).

Up to 23% of women were lost in follow up (Table 1). Between 21 and 32% of women in the intervention arms were lost in follow up, compared to only 15% in the control arm (*p* = 0.23). Furthermore, women in the ISF supplement and diet intervention arms were more likely to be lost due to adverse events (61.5% and 50.0% vs. 16.7%), whereas in the control arm, most women were lost due to the COVID-19 pandemic (50.0% vs. 23.1% and 12.5%). However, these differences were not statistically significant.

Importantly, the women who completed the study were similar to those who were lost in follow up (Appendix A). This suggests that failure to follow up did not significantly impact randomization. However, it is important to note that there were differences in MD at enrolment between those who were lost in follow up and those who completed the study for both dense area (19.2 cm^2^ vs. 14.6 cm^2^, *p* = 0.07) and percent density (21.0% vs. 12.7%, *p* = 0.03).

### 3.2. Description of Participants in the Primary Analysis

A total of 90 women were included in the primary analysis (Figure 1). As described in Table 2, women were 57 years old on average and were mostly of Chinese ethnicity (78.9%). Most women completed tertiary education (67.8%), and there was equal distribution by income category. Almost half of women reported that their last menstrual period was within 5 years of enrolment into the study (41.1%). However, women in the intervention arms reported a later age at last menstrual period compared to women in the control arm, though this was not statistically significant (51–52 years old vs. 49 years old, *p* = 0.08). Apart from this, all other patient characteristics were evenly distributed across study arms.

Twelve percent of women reported at least one first degree family member with breast cancer. For many women (65.6%), their last mammogram was more than 2 years ago. Four women reported that this was their first mammogram. Forty-one percent of women were obese or overweight at enrolment, and most women (51.1%) reported low levels of physical activity. These risk factors were evenly distributed across study arms.

Notably, women in the ISF supplement arm had higher MD measures for both absolute density (19.9 cm^2^ vs. 11.3 cm^2^ and 7.6 cm^2^, *p* = 0.17) and percent density (14.5% vs. 9.0% and 7.8%, *p* = 0.20). However, the differences across study arms were not statistically significant.

We also assessed for changes in body measurements and diet over the study period (Table 3). We observed no significant changes in weight, BMI, or waist-to-hip ratio across study arms. As expected, there were significant dietary changes. The increase in total calorie intake in the ISF diet arm, from 1586 to 1789 kcal/day (*p* = 0.10), likely corresponded to the increase in protein intake (by 16.5 g/day, *p* = 0.006) and fat intake (by 7.4 g/day, *p* = 0.04) due to the dietary soy intervention. In the ISF supplement arm, we observed a decrease in total calorie intake, from 1528 to 1496 kcal/day (*p* = 0.01), due to a decrease in protein intake (*p* = 0.04) and fat intake (*p* = 0.02). We observed no significant dietary changes for women in the control arm.

There were subtle differences in soy isoflavone intake at enrolment (Table 3). Women in the ISF Supplement arm reported higher habitual isoflavone intake compared to women in the ISF diet and control arms (27.9 mg vs. 17.6 mg and 15.3 mg per day, *p* = 0.12), but the difference across arms is small and not statistically significant. As expected, soy isoflavone intake increased significantly for women in the ISF supplement arm (from 27.9 mg/day to 101.4 mg/day, *p* < 0.01) and ISF diet arm (from 17.6 mg/day to 82.3 mg/day, *p* < 0.01) but not in the control arm (from 15.3 mg/day to 18.1 mg/day, *p* = 0.06). The main food sources of soy isoflavones consumed in the study were soymilk, soybean curd, and tofu.

### 3.3. Change in Mammographic Density over the Study Period

Women in the ISF Supplement arm appeared to have a larger absolute decline in dense area over time, by 1.3 cm^2^ compared to 0.5 cm^2^ in the ISF diet arm and 0.8 cm^2^ in the control arm, but the difference was small and not statistically significant (*p* = 0.48, Table 4). There were no significant differences by relative change in dense area (*p* = 0.99). Similarly, no statistically significant associations were noted for percent density measures (Appendix A) as well as Volpara^TM^ dense volume measures (Appendix A).

We conducted several sensitivity analyses to test the robustness of our observations. First, limiting the analysis to those who were compliant to the intervention (per-protocol analysis) did not improve the associations for both absolute change (*p* = 0.66) and relative change (*p* = 0.94), nor did excluding women with high isoflavone intake at enrolment. Second, based on the observation that women with low MD were most likely to observe the small-to-no absolute change (Appendix A), we excluded women with low MD at enrolment. This did not affect the observed differences in absolute MD change across study arms but resulted in a more pronounced difference for relative MD change (*p* = 0.37). Finally, we grouped women based on their total isoflavone intake (from diet and supplements) reported at Month 12. We found that the greatest absolute decline in MD was observed among women with moderate isoflavone intake (18–61 mg/day), followed by women with high isoflavone intake (61–101 mg/day) per day. There was little change in MD for women with very high isoflavone intake (>101 mg/day). On the contrary, the greatest relative change was observed for women with the lowest isoflavone intake compared to higher intake (−20.7% vs. 7–11%, *p* = 0.61). However, these observations were not statistically significant. It is important to note that up to 43% of women in the control arm reported isoflavone intake of between 18 and 61 mg/day (Appendix A).

We further stratified the analysis by time since menopause at enrolment. For women with a shorter duration since menopause (<5 years), there was a large absolute decline in mammographic density among women in the ISF supplement arm (−5.9 cm^2^ vs. −1.1 cm^2^ in the ISF diet arm and −0.8 cm^2^ in the control arm, *p* = 0.13, Figure 2a). We also observed a dose-response relationship among these women when assessed by quartiles of total isoflavone intake, where the largest effect was observed for women with highest isoflavone intake (Figure 2b). However, these associations were based on a small sample size and were not statistically significant. For women who reported more than 5 years since menopause, on the other hand, we observed little change in MD and no differences between study arms (Figure 2a). In contrast to women with a shorter duration since menopause, women who underwent menopause more than 5 years ago observed an increase in MD with very high isoflavone intake (>101 mg/day, *p* = 0.08, Figure 2b).

### 3.4. Changes in Mammographic Density over Time

We observed that the decline in MD prior to the intervention, over an average of 5–7 years, appeared to be greater for women in the control arm (−9.4 cm^2^) compared to women in the ISF diet arm (−8.3 cm^2^) and ISF supplement arm (−6.5 cm^2^, *p* = 0.47, Table 5). However, the opposite was observed during the 1-year study period, where the largest decline was observed among the ISF supplement arm (−1.9 cm^2^) followed by the ISF diet arm (−0.7 cm^2^), with little change in the control arm (0.1 cm^2^, *p* = 0.23).

### 3.5. Adverse Events

Of the 118 participants who were enrolled and randomized, 62.7% reported at least one adverse event during the study period (Table 6). In this unblinded study, adverse events were most reported among women in the intervention arms (78–82% vs. 28% in the control arm, *p* < 0.001). Common complaints include gastrointestinal problems (19.5%, *p* < 0.001), skin rash (10.2%, *p* = 0.07), joint pains (7.6%, *p* = 0.06), weight gain (6.8%, *p* = 0.17), and numbness (2.5%, *p* = 0.11). Most of these complaints were more common with higher dietary soy intake, except for joint pain and numbness, which were more common with supplement intake. There were also a few serious adverse events in the ISF supplement arm over the study period, including one case of post-menopausal bleeding, two cases of ruptured brain aneurysms, and a breast cancer diagnosis.

## 4. Discussion

In a randomized controlled trial of 90 healthy peri- and postmenopausal Malaysian women, we found no statistically significant differences in annual MD change for women with or without the soy isoflavone intervention. However, we found that high soy isoflavone intake may benefit women who are perimenopausal or recently menopausal, where an annual decline of up to 6 cm^2^ was observed among women on the 100 mg/day ISF supplement compared to the 1 cm^2^ decline in the control arm.

We demonstrated a possible modest association between regular soy isoflavone intake and MD change among Asian women. These observations were substantiated in within-women analysis prior to the intervention for a subset of 62 women for whom a pre-enrolment mammogram was available. Despite some inconsistencies in observational studies [34,35], MD change is reported to be a good proxy for breast cancer risk in clinical trials of endocrine therapy response [36]. For example, studies of tamoxifen for primary prevention of breast cancer have shown that a 10% decline in percent density over a period of 1.5 years was associated with a 63% lower relative risk of breast cancer [37]. However, little is known about the impact of more subtle changes in MD, such as that observed in this study.

The association between moderate soy isoflavone intake and MD in our study is consistent with the published literature among Asian women, where habitual intake of at least 10 mg/day of isoflavones or 10 g/day of soy foods was associated with 10–16% lower relative risk of breast cancer [13,38,39]. In other studies, doses of up to 40 mg/day were required to show a protective effect [40]. In our study, however, the moderate intake of soy isoflavones among women in the control arm likely resulted in smaller effect sizes observed for women in intervention arms. Nevertheless, the findings here suggest that with sufficient sample size and an appropriate control arm, a causal association between moderate soy isoflavone intake and breast cancer risk might be established among Asian women.

Our findings are in contrast to previous clinical trials of predominantly Caucasian women, where intervention doses of up to 120 mg/day of soy isoflavones have yielded no significant impact on MD measures [16,17,18,19,20,21]. The lack of association in these studies was attributed to the use of soy supplements rather than soy foods made from whole soy beans [22]. Soy foods may contain other nutritional components that may increase the potency of soy isoflavones, and they tend to be consumed several times over the day, resulting in continuous exposure [16,19,41]. By having both a high dose supplement arm and a moderate dose soy food arm in our study, we postulate that the effect of soy isoflavones on mammographic density does not depend on how the isoflavones are consumed, either through supplements or dietary intake.

Researchers have also suggested that the lack of effect seen in clinical trials of post-menopausal women [16,17,18,19,20,21] is because it misses the critical window for soy intake, which is hypothesized to be during adolescence [42,43,44]. Here, we show that the timing around or soon after menopause could be a critical window for soy isoflavone intake. Similar observations were noted recently in a low-dose tamoxifen study, where a protective effect on MD was only observed among older pre-menopausal women but not among post-menopausal women [45]. As women experience menopause, the breast undergoes a process of lobular involution, where the fibro-glandular tissue (radiologically dense tissue) progressively transforms into fat (radiologically non-dense tissue) [46]. Therefore, women naturally experience the greatest MD decline over the menopausal transition [47]. Furthermore, higher rates of decline during this time were shown to be associated with reduced breast cancer risk [48,49]. This supports our hypothesis that high soy isoflavone intake among women around the time of menopause could meaningfully reduce post-menopausal breast cancer risk. Conversely, we demonstrate that very high isoflavone intake long after menopause may be harmful to breast health. However, due to the small sample sizes in our study, these findings will require confirmation in a larger RCT that is powered to investigate differences by more refined intervals of time since menopause to better understand the window of susceptibility for soy intake.

This is the first RCT to investigate the effect of soy isoflavones on breast cancer risk among Asian women living in Asia, where soy is commonly available and is already part of dietary intake. To our knowledge, we are also the first to show that a soy intervention may specifically benefit women during or soon after menopause, possibly coinciding with the lobular involution of the breasts. Further investigation of this hypothesis may provide insight into the biological mechanism of how soy isoflavones affect breast cancer risk. An important strength of this trial is that all mammograms were conducted on a single digital mammography unit, and mammograms for each woman was aligned prior to MD estimation, thereby ensuring a robust measurement of MD change for each woman.

This trial had some limitations. First, we were only able to enroll 52% of our desired sample size, and therefore, this study is underpowered to detect any meaningful difference in MD change across study arms. Additionally, up to 23% of women were lost in follow up. Though the rate of loss is comparable to previously published dietary intervention studies [50], this loss was not at random in our study and limited our ability to conduct intention-to-treat analyses. Poor recruitment and failure to follow up were further exacerbated by the COVID-19 pandemic and corresponding national lockdowns during the study period. These events may have also led to periods of poor compliance. Second, despite our best efforts to recruit women with low soy intake, some women in the control arm reported moderate intake of soy isoflavones, which may have attenuated any real effect of the study intervention on MD in this study. Third, the use of self-reported measures of intake of soy supplements or dietary soy may have led to exposure misclassification. Finally, many women experiencing the isoflavone interventions reported adverse events, though most reported mild symptoms. Notably, there were several instances of serious adverse events among women in the ISF supplement arm. Previous studies of up to 120 mg/day of soy isoflavones from supplements have reported no serious events [15], and therefore, our findings are likely due to chance. Nonetheless, a more robust assessment of the safe and effective dosing range for soy isoflavones is required.

## 5. Conclusions

We demonstrate a plausible causal association between soy isoflavone intake and mammographic density change, as a biomarker of breast cancer risk, among Asian women around or soon after menopause. However, due to the limited sample size in this trial, the findings should be interpreted with caution and require confirmation in a larger trial of Asian women that appropriately accounts for the habitual intake of soy isoflavones in this population. Nonetheless, it suggests that a diet rich in soy isoflavones could be an important, cost-effective, and acceptable primary prevention strategy for breast cancer risk in a region that is currently observing alarming increases in breast cancer incidence.

## Figures and Tables

**Figure 1 nutrients-15-00299-f001:**
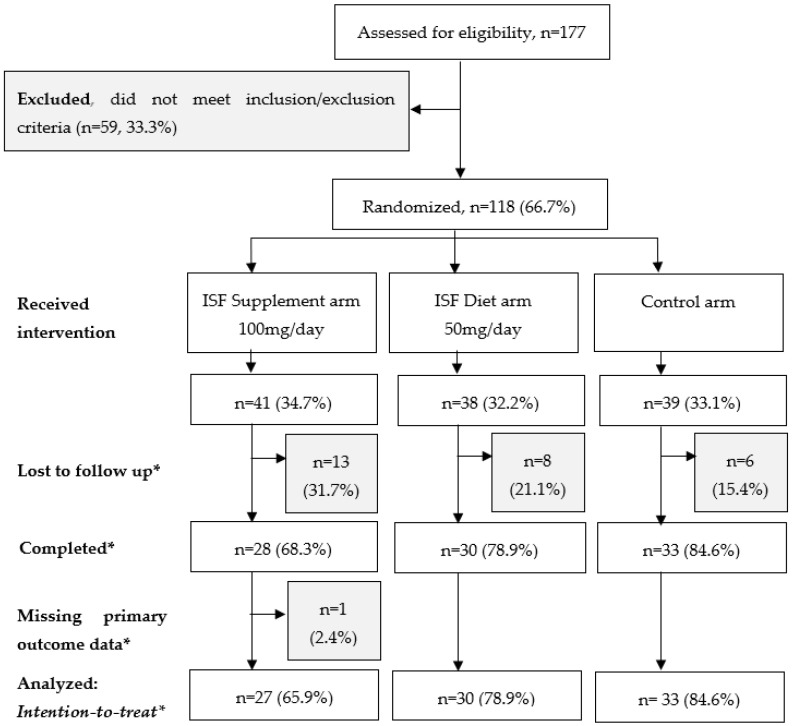
CONSORT flow diagram for participant enrolment and follow up. * The proportions reported are calculated with the number of women who received allocated intervention as the denominator.

**Figure 2 nutrients-15-00299-f002:**
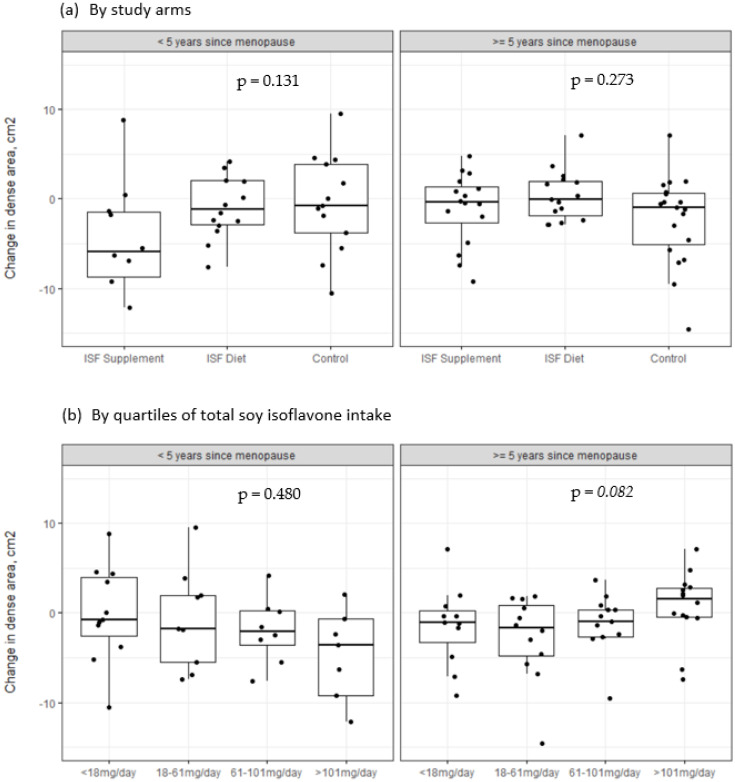
Distribution of absolute change in dense area by time since menopause using (**a**) study arm and (**b**) quartiles of total reported soy isoflavone intake (including supplements and diet).

**Table 1 nutrients-15-00299-t001:** Follow up among participants who received allocated intervention.

		Distribution of Participants by Study Arm			
Description of Follow Up	Overall (*n* = 118)	ISF Supplements (*n* = 41)	ISF Diet (*n* = 38)	Control (*n* = 39)	*p* Value
Follow up status, *n* (%)					
Completed	91 (77.1)	28 (68.3)	30 (78.9)	33 (84.6)	0.225
Lost in follow up	27 (22.9)	13 (31.7)	8 (21.1)	6 (15.4)	
Duration of follow up, months, median (IQR)					
Completed	13.3 (1.4)	13.5 (1.7)	13.2 (1.3)	13.2 (1.2)	0.651
Lost in follow up	6.2 (8.8)	6.4 (9.3)	4.1 (5.9)	8.1 (2.6)	0.354
Reasons for failure to follow up, *n* (%)					
Adverse events	13 (48.1)	8 (61.5)	4 (50.0)	1 (16.7)	0.332
No longer interested	7 (25.9)	2 (15.4)	3 (37.5)	2 (33.3)	
COVID-19 pandemic	7 (25.9)	3 (23.1)	1 (12.5)	3 (50.0)	

**Table 2 nutrients-15-00299-t002:** Distribution of completed participants (*n* = 90) by study arm.

		Distribution by Study Arm			
Characteristics	Overall (*n* = 90)	ISF Supplements (*n* = 27)	ISF Diet (*n* = 30)	Control (*n* = 33)	*p* Value
Demographic					
Age in years, median (IQR)	57 (6.0)	58 (5.0)	56 (5.8)	56 (8.0)	0.470
Ethnicity, *n* (%)					
Chinese	71 (78.9)	20 (74.1)	25 (83.3)	26 (78.8)	0.781
Indian	11 (12.2)	5 (18.5)	2 (6.7)	4 (12.1)	
Malay	8 (8.9)	2 (7.4)	3 (10.0)	3 (9.1)	
Education, *n* (%)					
Up to secondary	28 (31.1)	8 (29.6)	12 (40.0)	8 (24.2)	0.410
Tertiary	61 (67.8)	18 (66.7)	18 (60.0)	25 (75.8)	
Monthly household income, *n* (%)					
<RM 5000	33 (36.7)	10 (37.0)	12 (40.0)	11 (33.3)	0.987
RM 5000–10,000	28 (31.1)	8 (29.6)	9 (30.0)	11 (33.3)	
>RM 10,000	26 (28.9)	8 (29.6)	8 (26.7)	10 (30.3)	
Reproductive					
No. of children, *n* (%)					
None	17 (18.9)	5 (18.5)	7 (23.3)	5 (15.2)	0.746
1–2	40 (44.4)	11 (40.7)	15 (50.0)	14 (42.4)	
>3	32 (35.6)	10 (37.0)	8 (26.7)	14 (42.4)	
Menopausal status at enrolment, *n* (%)					
Peri-menopause	9 (10.0)	4 (14.8)	2 (6.7)	3 (9.1)	0.618
Post-menopause	81 (90.0)	23 (85.2)	28 (93.3)	30 (90.9)	
Age at last menstrual period, median (IQR)	51 (5.5)	52 (4.0)	51 (5.0)	49 (5.0)	0.079
Years since last menstrual period, *n* (%)					
Less than 5 years	37 (41.1)	10 (37.0)	14 (46.7)	13 (39.4)	0.736
5 years or more	50 (55.6)	16 (59.3)	15 (50.0)	19 (57.6)	
History of hysterectomy or oophorectomy, *n* (%)	10 (11.1)	2 (7.4)	2 (6.7)	6 (18.2)	0.310
Use of oral contraceptives, *n* (%)	29 (32.2)	8 (29.6)	11 (36.7)	10 (30.3)	0.847
Family history and screening					
1° Family history, *n* (%)					
Any cancer	42 (46.7)	16 (59.3)	13 (43.3)	13 (39.4)	0.289
Breast cancer	11 (12.2)	5 (18.5)	3 (10.0)	3 (9.1)	0.475
Last mammogram, *n* (%)					
1–2 years	27 (30.0)	8 (29.6)	10 (33.3)	9 (27.3)	0.405
More than 2 years	59 (65.6)	18 (66.7)	17 (56.7)	24 (72.7)	
Never	4 (4.4)	1 (3.7)	3 (10.0)	0 (0)	
Lifestyle factors at enrolment					
BMI, kg/m^2^, *n* (%)					
<25 (low)	53 (58.9)	15 (55.6)	17 (56.7)	21 (63.6)	0.800
≥25 (high)	37 (41.1)	12 (44.4)	13 (43.3)	12 (36.4)	
Physical activity, MET-hours/week, *n* (%)					
≤10 (low)	46 (51.1)	16 (59.3)	13 (43.3)	17 (51.5)	0.663
>10 (moderate/high)	41 (45.6)	11 (40.7)	15 (50.0)	15 (45.5)	
Mammographic density at enrolment, median (IQR)					
Dense area, cm^2^	14.6 (18.7)	19.9 (15.0)	11.3 (19.3)	7.6 (18.1)	0.172
Percent density, %	12.7 (22.1)	14.5 (21.4)	9.0 (21.0)	7.8 (22.4)	0.200

**Table 3 nutrients-15-00299-t003:** Changes to body measurements and dietary intake over the intervention period (*n* = 90).

	Distribution by Study Arm, Median (IQR)			
Measures	ISF Supplements (*n* = 27)	ISF Diet (*n* = 30)	Control (*n* = 33)	p_arm_
Weight, kg				
At enrolment	61.2 (11.8)	59.1 (12.9)	58.5 (13.3)	0.931
After 12 months	61.5 (15.5)	60.9 (14.7)	58.8 (14.2)	0.956
p_change_	0.999	0.524	0.290	
BMI, kg/m^2^				
At enrolment	24.2 (5.2)	24.0 (4.2)	23.0 (5.5)	0.941
After 12 months	24.2 (5.7)	24.3 (5.8)	24.0 (5.3)	0.967
p_change_	0.878	0.503	0.272	
Waist-to-hip ratio				
At enrolment	0.9 (0.1)	0.9 (0.1)	0.9 (0.1)	0.545
After 12 months	0.8 (0.1)	0.9 (0.1)	0.9 (0.1)	0.834
p_change_	0.934	0.598	0.177	
Calorie intake, kcal/day				
At enrolment	1528.1 (333.2)	1586.5 (686.1)	1528.7 (727.1)	0.816
After 12 months	1496.2 (525.1)	1788.9 (692.8)	1556.4 (848.4)	0.200
p_change_	0.012	0.097	0.708	
Carbohydrate intake, g/day				
At enrolment	223.4 (91.9)	239.8 (137.9)	232.6 (132.2)	0.946
After 12 months	228.1 (93.2)	257.6 (109.8)	242.8 (122.5)	0.382
p_change_	0.097	0.330	0.881	
Protein intake, g/day				
At enrolment	58.3 (22.2)	54.4 (23.6)	56.7 (30.0)	0.424
After 12 months	51.5 (29.2)	70.9 (23.6)	59.6 (25.7)	0.004
p_change_	0.044	0.006	0.408	
Fat, g/day				
At enrolment	43.0 (20.4)	37.0 (20.5)	39.3 (24.3)	0.084
After 12 months	42.5 (17.2)	44.4 (22.9)	39.3 (33.3)	0.830
p_change_	0.021	0.040	0.315	
Soy isoflavone intake, mg/day				
At enrolment	27.9 (25.6)	17.6 (26.4)	15.3 (17.2)	0.121
After 12 months	101.4 (55.6)	82.3 (39.1)	18.1 (19.2)	<0.001
p_change_	<0.001	<0.001	0.062	

**Table 4 nutrients-15-00299-t004:** Change in dense area during the study period, by study arm (*n* = 90).

		Change in Dense Area, Median (IQR)			
Study arm, by Type of Analysis	n	Absolute Change (cm^2^)	*p* Value	Relative Change (%)	*p* Value
All completed participants					
ISF Supplement	27	−1.3 (7.3)	0.479	−11.5 (43.8)	0.999
ISF Diet	30	−0.5 (4.5)		−10.9 (48.1)	
Control	33	−0.8 (6.3)		−17.8 (51.6)	
Per-protocol participants ^†^					
ISF Supplement	15	−0.5 (8.4)	0.659	−11.1 (51.5)	0.935
ISF Diet	24	−0.5 (4.5)		−10.9 (47.6)	
Control	29	−0.6 (6.3)		−17.8 (54.0)	
Sensitivity analyses					
Excl. low MD at enrolment ^‡^					
ISF Supplement	23	−1.7 (7.6)	0.449	−11.5 (35.5)	0.372
ISF Diet	20	−0.9 (5.0)		−2.3 (28.9)	
Control	21	−1.0 (8.8)		−6.9 (47.4)	
Excl. high enrolment isoflavone intake ^§^					
ISF Supplement	16	−0.9 (6.2)	0.573	−10.3 (37.4)	0.908
ISF Diet	20	−0.5 (4.7)		−7.9 (49.8)	
Control	29	−0.8 (6.3)		−17.8 (50.3)	
By total isoflavone intake					
<18 mg/day	23	−0.8 (5.5)	0.555	−20.7 (55.8)	0.610
18–61 mg/day	22	−1.8 (6.9)		−11.9 (56.0)	
61–101 mg/day	21	−1.3 (3.2)		−9.4 (34.9)	
>101 mg/day	24	−0.2 (6.9)		−7.4 (50.1)	

^†^ Reporting >80% compliance at end of study. ^‡^ Excluding the first 25th percentile of the distribution for mammographic density (<5.5 cm^2^). ^§^ Excluding the last 25th percentile of the distribution (>32.7 mg/day for isoflavone intake at baseline).

**Table 5 nutrients-15-00299-t005:** Change in dense area over time (*n* = 62).

Description of Previous Mammogram and Change over Time	Distribution of MD Change by Study Arm			
	ISF Supplements (*n* = 19)	ISF Diet (*n* = 21)	Control (*n* = 22)	*p* Value
Duration since previous mammogram in years, median (IQR)	5.4 (2.1)	6.7 (1.5)	6.0 (2.1)	0.103
Menopause status at previous mammogram, *n* (%)				
Pre-menopause	8 (42.1)	4 (19.0)	8 (36.4)	0.593
Peri-menopause	3 (15.8)	5 (23.8)	4 (18.2)	
Post-menopause	7 (36.8)	11 (52.4)	9 (40.9)	
Dense area in previous mammogram, cm^2^, median (IQR)	29.3 (24.3)	32.6 (27.5)	33.7 (31.0)	0.901
Absolute change in dense area (cm^2^), median (IQR)				
Change prior to intervention	−6.5 (8.4)	−8.3 (14.6)	−9.4 (11.7)	0.470
Change during intervention	−1.9 (7.4)	−0.7 (5.0)	0.1 (6.8)	0.225
Relative change in dense area (%), median (IQR)				
Change prior to intervention	−11.8 (57.9)	−39.7 (31.8)	−33.0 (30.3)	0.292
Change during intervention	−15.5 (35.4)	−7.4 (34.7)	−2.5 (32.9)	0.460

**Table 6 nutrients-15-00299-t006:** List of reported adverse events among participants who received allocated intervention.

		Distribution of Participants by Study Arm			
Adverse Events	Overall (*n* = 118)	ISF Supplements (*n* = 41)	ISF Diet (*n* = 38)	Control (*n* = 39)	*p* Value
Overall	74 (62.7)	32 (78.0)	31 (81.6)	11 (28.2)	<0.001
Probably related					
Gastrointestinal complaints	23 (19.5)	10 (24.4)	12 (31.6)	1 (2.6)	0.001
Skin rash	12 (10.2)	4 (9.8)	7 (18.4)	1 (2.6)	0.070
Joint pains	9 (7.6)	5 (12.2)	4 (10.5)	0 (0)	0.062
Weight gain	8 (6.8)	2 (4.9)	5 (13.2)	1 (2.6)	0.169
Numbness	3 (2.5)	3 (7.3)	0 (0)	0 (0)	0.106
Uncertain					
High blood uric acid	4 (3.4)	3 (7.3)	1 (2.6)	0 (0)	0.268
Back pain	4 (3.4)	3 (7.3)	1 (2.6)	0 (0)	0.268
Headache	3 (2.5)	1 (2.4)	2 (5.3)	0 (0)	0.422
High blood glucose	2 (1.7)	0 (0)	1 (2.6)	1 (2.6)	0.542
High liver function tests	2 (1.7)	2 (4.9)	0 (0)	0 (0)	0.328
Changes to the breast	2 (1.7)	0 (0)	1 (2.6)	1 (2.6)	0.543
Vaginal discharge	2 (1.7)	2 (4.9)	0 (0)	0 (0)	0.328
Sleep disturbance	2 (1.7)	2 (4.9)	0 (0)	0 (0)	0.328
Uncertain, serious					
Brain aneurysm	2 (1.7)	2 (4.9)	0 (0)	0 (0)	0.328
Breast cancer	1 (0.8)	1 (2.4)	0 (0)	0 (0)	0.322
Post-menopausal bleeding	1 (0.8)	0 (0)	1 (2.6)	0 (0)	0.322
Unrelated					
Infection	26 (22.0)	10 (24.4)	9 (23.7)	7 (17.9)	0.788
Surgery	6 (5.1)	2 (4.9)	2 (5.3)	2 (5.1)	0.999
High blood pressure	5 (4.2)	2 (4.9)	1 (2.6)	2 (5.1)	0.999
Injury/accident	3 (2.5)	1 (2.4)	1 (2.6)	1 (2.6)	0.999

## Data Availability

Data can be made available upon reasonable request.

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
