# Peer review of "A Randomized Controlled Trial of Soy Isoflavone Intake on Mammographic Density among Malaysian Women"

_nutrients, 2023, doi:10.3390/nu15020299_

Round 1

Reviewer 1 Report

Rajaram and colleagues conducted a randomized controlled trial of soy isoflavone intake on mammographic density among Malaysian women.   My main comments are as follows:

Tables 2 and 3   Baseline intake of soy in the 3 groups (27 ISF supplement, 30 ISF diet, and 33 control) was presented in Table 2.  However, soy intake in the 3 groups after 12 months was not shown in Table 3 but was shown in Supplemental Table 3 as a categorical variable (<18, 18-61, 61-101, >101 mg/day).  It is important to include soy intake at baseline and after 12 months in Table 3, showing both median of intake (IQR) as well as in a categorical variable. 
In Table 2- median age at menstrual period was presented-  please clarify whether all the women achieve menopause naturally.  Please give the numbers if some women had hysterectomy or oophorectomy.  

Table 5  Change in dense area over time in 62 participants who had mammograms prior to intervention was shown. Since the median years ranged from 5.4 (ISF supplement) to 6.2 (ISF-diet) groups, presumably most of them were premenopausal at the time of previous mammogram.  Given the large change in mammographic density when women transition from  premenopausal to postmenopausal status, interpretation of these results would require knowing if menopausal status has changed in these women.  In addition, the information we have on baseline density (from Table 2) was based on 90 women.  Among subjects shown in Table 5, how many were pre, peri  vs postmenopausal at previous mammogram, and what was the absolute dense area for this subgroup of women (since information presented in Table 2 were for all 90 women). 

An important paper by Martin LJ  et al. (Breast Cancer Res Treatment 2009, 113: 163-172) highlighted the challenges of studying changes in mammographic density among women in relation to dietary fat intervention among women in the premenopausal-postmenopausal transition.   Hence, it is important to be cautious in interpreting the more substantial reduction among those in the <5 years since menopause group.  Even if your numbers are too small among the <5 years group, it would be informative to know how many were perimenopausal in your study.  Future studies with larger sample sizes would need to refine this to <1-2 years, 3-5 years since menopause to better understand the utility of this ‘window’ of susceptibility for   interventions such as soy or other agents.

5.  Sections 2.2 and 2.3  What were the main soy foods consumed by participants in Malaysia? Subjects were not ‘given’ soy foods but a cash supplement to purchase soy foods.

6.  Section 3.1, lines 198-201 and Supplementary Table 1   The differences in MD between those who were lost to follow-up compared to those who completed the study seemed quite substantial instead of small.  Also clarify in that 19.2 cm2 was for those lost to follow-up and 14.6 cm2 was for those who completed. Similarly, 21.0% was for those who were lost to follow-up and 12.7% was for those who completed the study.

7.  The authors concluded that unlike previous studies (Ref 13-19), this study would suggest that the effect of soy isoflavones on mammographic density does not depend on how the isoflavone are consumed either through supplements of dietary intake.  The overall results shown in Table 4 for all completed participants and per protocol participants were not dissimilar with previous studies.   It should be noted that the lack of subjective assessment of soy intake is an important limitation of this study.

Reviewer 2 Report

The paper is focused on a Randomized Controlled Trial of Soy Isoflavone Intake on Mammographic Density among Malaysian Women. Extensive revisinn of the manuscript is needed. Please see below my main suggestions regarding this research:

1.     ZIP code must be added for each affiliation.

2.     References must be cited in the text in the MDPI style [ ], not as superscript. Please see the Instructions for authors https://www.mdpi.com/journal/nutrients/instructions .

3.     L46. Why “in this region”? develop the paragraph better.  Literature specify many results of soy isoflavone intake as hormone replacement therapy, I suggest checking and referring to https://doi.org/10.3390/jcm7100297 and PMID: 29167771. Moreover, dietary interventions like phytonutrients must be referred to – please see  https://doi.org/10.3390/biom11081176

4.     Aim of the study, L72-75. Please, develop it. Which is the novelty of this paper? What special aspects brings to the field? Why have you chosen this topic? Detail your reasons.

5.     Figure 1 is blurred, please replace it with a better quality one.

6.     L113. Regarding the questionnaires, few details must be added:

-       How did you choose the items? You should motivate each item.

-       Did you classify them? 

-       Which list did you considered?

-       You should introduce the method used, why it is useful for your investigation, what is the objective of your study related to the questionnaire and the main impacts that you want to obtain after the data gathered.

-       How were the sample chosen? Is it representative of the population in the area/country?

-       Was the questionnaire pretested?

-       Have you worked with a sociologist in making the questionnaire?

-       Who validated the questionnaire?

7.     L126. What does “assessed regularly” mean, at what interval were they checked?

8.     L131. Producer of the supplement must be mentioned, as well.

9.     Complete the head of the Table 1 and 4 above the 1st column (what represents the first column?)

10.  Head of Tables 1, 2 ,3 and 6. Please complete in the head of the table the explanation for the values written in parenthesis in the columns (I consider %)

11.  Table 2 must be merged, not in 2 parts.

12.  Remove empty lines, 237, 238, 278, 279, etc. Aspect of the paper has same importance as the content.

13.  Table 4. Explain in the head of the table the meaning of the values in the parenthesis, in the columns 3 and 5.

14.  References must be written in the MDPI style. See the Instructions for authors.

Round 2

Reviewer 2 Report

The authors made the requested improvements.